# Population Dynamics of *Phytophthora infestans* in Egypt Reveals Clonal Dominance of 23_A1 and Displacement of 13_A2 Clonal Lineage

**DOI:** 10.3390/jof9030349

**Published:** 2023-03-13

**Authors:** Sherif Mohamed El-Ganainy, Ahmed Mahmoud Ismail, Maali Shaker Soliman, Yosra Ahmed, Muhammad Naeem Sattar, Biju Vadakkemukadiyil Chellappan, David E. L. Cooke

**Affiliations:** 1Department of Arid Land Agriculture, College of Agricultural and Food Sciences, King Faisal University, P.O. Box 420, Al-Ahsa 31982, Saudi Arabia; amismail@kfu.edu.sa; 2Pests and Plant Diseases Unit, College of Agricultural and Food Sciences, King Faisal University, P.O. Box 420, Al-Ahsa 31982, Saudi Arabia; 3Plant Pathology Research Institute, Agricultural Research Center, Giza 12619, Egypt; maali_soliman@yahoo.com (M.S.S.); yosra.ahmed@fao.org (Y.A.); 4The Central Laboratory for Phytosanitary and Food Safety, United Integrated Laboratories, Barka 662, Oman; 5Regional Office for Near East and North Africa, Food and Agriculture Organization of the United Nations, Dokki 2223, Egypt; 6Central Laboratories, P.O. Box 420, Al-Ahsa 31982, Saudi Arabia; mnsattar@kfu.edu.sa; 7Department of Biological Sciences, College of Science, King Faisal University, P.O. Box 420, Al-Ahsa 31982, Saudi Arabia; bchellappan@kfu.edu.sa; 8The James Hutton Institute, Invergowrie, Dundee DD2 5DA, UK

**Keywords:** genotyping, multi locus alignments, late blight, mating type, AVR2, eggplant

## Abstract

Potato (*Solanum tuberosum* L.) and tomato (*S. lycopersicum* L.) are the most economically important vegetable crops in Egypt and worldwide. The winter crop in Egypt is particularly prone to late blight caused by *Phytophthora infestans*. A total of 152 *P. infestans* isolates were isolated from the 2013, 2014, 2016 and 2018 winter crops with 82 isolates from potato, 69 from tomato and one isolate from eggplant (*S. melongena* L.). All isolates belonged to the A1 mating type with no evidence of A2 or self-fertile strains. The majority of isolates (53%) were sensitive to metalaxyl, 32% were intermediate and 15% were resistant. Variation in aggressiveness between three *P. infestans* isolates EG-005 (13_A2) and EG-276 (23_A1) from potato, and EG-237 (23_A1) from eggplant was determined on tuber slices and leaflets of 10 potato cultivars. The eggplant isolate EG-237 showed higher sporulation capacity compared with the other tested isolates and was able to infect potato (Lady Rosetta cv) and tomato (Super Strain B cv). The simple sequence repeat (SSR) genotyping data showed that in contrast to our previous work (3-year period 2010–12) in which the proportion of 13_A2 lineage was 35%, all isolates belonged to the 23_A1 lineage. There was no evidence for the existence of the A2 mating type or 13_A2 lineage even in the destroyed field crops of some cultivars (Cara, Bellini and Valor) that had been reported as resistant to 23_A1. The data have been submitted into the Euroblight database to allow temporal and spatial genetic diversity to be examined in comparison with other regional *P. infestans* populations. The AVR2 and AVR2-like RXLR effector genes were amplified and sequenced. In the avirulent AVR2 gene, only one heterozygous SNP was detected at position 31 in the N terminus in six isolates out of eleven, whereas two heterozygous SNPs were detected at position 29 in the N-terminus and ninety-two in the C- terminus of the AVR2-like gene. This suggests that changes in the previously reported virulence profile of 23_A1 are not related to commercial cultivars carrying the R2 gene. In addition, this is the first report of *P. infestans* on eggplant in Egypt.

## 1. Introduction

*Phytophthora infestans*, the causal pathogen of late blight, is a globally important and long-studied pathogen of potato (*Solanum tuberosum* L.) and tomato (*S. lycopersicum* L.) [1,2]. It has been claimed as the major factor of the infamous 1846 Irish famine and is still a major cause of heavy yield losses to potato production worldwide [3]. The algal-like stramenopile pathogen *P. infestans* is a heterothallic and hemibiotrophic oomycete with A1 and A2 mating types that, if in close proximity, form oospores, enabling it to reproduce sexually in addition to the asexual sporangia. Some of the key features that differentiate oomycetes from the true fungi are the presence of β-1,3-glucan and cellulose in the cell wall composition, diploidy during most of the life cycle, production of motile spores, coenocytic mycelia, and production of gametangia [4]. The co-existence of both mating types is required for sexual recombination, which may lead to the production of stable and long-lasting soil-borne oospores [5]. In turn, the resulting sexual recombinants increase the genetic diversity of *P. infestans* and increase the threat of disease by providing an additional source for primary inoculum. The evolution of genetically more variable *P. infestans* isolates enhances the pathogen’s adaptability and epidemiological potential under diverse environments and on resistant crop genotypes [6]. In addition to sexual hybridization, long-distance dispersal of sporangia and transportation of infected plant material are key drivers of increased genetic diversity in the *P. infestans* population [7]. Therefore, studying the population structure and genetic diversity of *P. infestans* is important to unravel the evolutionary linkages, migration patterns, virulence capabilities and identify better management practices over time. Several phenotypic markers have been used for the characterization of the *P. infestans* population including mating type, metalaxyl sensitivity and the virulence profiles of the pathogen [8]. Moreover, for more accurate characterization, the genotypic markers including mitochondrial (mt) DNA haplotype, restriction fragment length polymorphisms (RFLPs) and simple sequence repeats (SSR) have been used [9,10]. For high-throughput *P. infestans* screening, a multiplex SSR-based one-step protocol is the most efficient method, which employs twelve SSR markers [10]. Besides understanding the above neutral markers, the study of effector diversity is also important [11]. Fungal pathogens suppress the host plant’s immunity by secreting a vast array of cytoplasmic and apoplastic effectors into the host cells. Two major cytoplasmic effectors in *P. infestans* include crinkling and necrosis proteins (CRN) and RxLRs. Most of the known avirulence (AVR) effector genes are of RxLR type, which is a unique class of host-translocated proteins consisting of an RxLR domain following a signal peptide at the N-terminal [12]. Thus, studying the repertoire of RxLR effectors may provide molecular insights into the interaction between emerging *P. infestans* lineages and the host plants with R-genes leading to the improved deployment of cultivar resistance for integrated management of late blight [13].

The only known mating type until the 1980s was the A1 mating type, which most likely migrated to Europe in 1840. Thus, sexual reproduction was not realized in *P. infestans* across many countries, except in Mexico, which is considered the center of origin of *P. infestans* with the earliest discovered co-existence of the A1 and A2 mating types [14]. The introduction of the A2 mating type to Europe occurred in 1980 in Germany with a claimed link to potato shipments from Mexico in 1976 [15,16,17].

The clonal lineage 13_A2 is a highly aggressive A2 mating type reported first in 2004 from the Netherlands and Germany [18]. After its subsequent appearances in Poland and the United Kingdom, it caused widespread epidemics in China and India, severely affecting potato and tomato crops [10]. In a long-term phenotypic survey in 2008–2014, the genotypes 13_A2, 2_A1 and 23_A1 overwhelmed the majority of the populations in Algeria [19]. In Africa, a *P. infestans* pandemic was first reported in Kenya in the early 1940s [20] and followed by its spread across the borders into the western parts of Congo and southern Tanzania. The *P. infestans* strain 2_A1 is a dominating A1 mating type in the east African countries Burundi, Kenya, Rwanda, Tanzania and Uganda [21]. Whereas, the population structure of *P. infestans* in the North African countries is mostly newly emerging A2 mating types of European origin [13,22]. This might be due to a high import of seed potato into this region from European countries [19].

Egypt is a significant exporter of potatoes with exports worth USD 221.9 million in 2020. *Phytophthora infestans* was presumably present in the agro-ecological regions of Egypt since 1941 [23] and some reports mentioned the presence of both the A1 and A2 mating types and additionally self-fertile isolates of *P. infestans* [13,24,25]. Although, in Egypt both mating types have been reported earlier. Similar results have been reported in Morocco, Tunisia and Algeria [19,26,27]. This sporadic distribution of diverse *P. infestans* lineages varied from year to year in Egypt [24,25,28,29]. Differences in the virulence profiles on differential potato clones (*Solanum demissum*) carrying the R genes (R1-R11), metalaxyl sensitivity and SSR markers have been used to characterize isolates of *P. infestans* in Egypt and have identified two European clonal lineages 13_A1 and 23_A1. Differences in virulence profiles were observed within the clonal lineages 13_A2 and 23_A1 [13].

In light of some observed differences in blight resistance breakdown, the present study investigated the population structure of *P. infestans* causing late blight epidemics in Egypt over the 2013 to 2018 seasons. Such a situation demands extensive genotyping and phenotyping data to help in developing better disease management approaches. Keeping this in view, the study was designed to: 1. Study the genetic diversity of various genotypes prevailing in potato and tomato crops in Egypt. 2. Understand the population structure of *P. infestans* in Egypt using SSR markers. 3. Analyze the virulence profile of *P. infestans* isolates on commercial cultivars and their metalaxyl sensitivity. 4. Determine the existing genetic diversity of the crucial effector genes.

## 2. Materials and Methods

### 2.1. Sample Collection and Isolation

Symptomatic potato and tomato plants showing late blight symptoms were collected from 37 locations belonging to 9 Egyptian governorates, i.e., Beheira, Dakahlia, Fayoum, Gharbia, Giza, Kafr El-Sheikh, Menofia, Qalyubia and Sharqia during fall and winter seasons of 2013, 2014, 2016 and 2018 (Figure 1A, Appendix A). The leaves and stem parts from the infected potato and tomato plants were collected and stored in plastic bags. The samples were washed under running tap water to remove debris and soil residues. All samples were blotted with tissue paper and air-dried. Next, the samples were placed in inverted Petri dishes containing water agar medium in the dark and incubated at 18 °C for 24 h. After sporulation appearance, the sporulated mycelium was picked and purified as described earlier [13]. The purified mycelial cultures were propagated and maintained on Rye slants and kept at 5 °C for further studies.

### 2.2. Mating Type and Metalaxyl Sensitivity Assessment

The mating type assay was performed as described previously. Briefly, the mycelial plugs from the sampled *P. infestans* isolates were placed 4 cm apart from the conventionally used A1 and A2 reference isolates on pea agar medium (PAM) and were grown together [13]. The plates were incubated at 18 °C in the dark for 14 days and finally evaluated for the presence or absence of oospores. The isolates showing oospore formation in the contact interface with the A1 or A2 mating type were designated as A2 and A1 isolates, respectively.

Metalaxyl sensitivity of the sampled 152 *P. infestans* isolates was tested using metalaxyl-amended rye agar medium (RAM) as described previously [13,30]. A 5 mm mycelial plug was picked from a 14-day-old colony and placed on RAM supplemented with 5 or 100 mg L^−1^ metalaxyl (32012 Sigma Aldrich). The same concentration of dimethyl sulfoxide (DMSO) was used to replace metalaxyl in the control plates. The experiment was performed with three technical replicates of the metalaxyl-amended plates incubated at 18 °C in the dark for the assessment of the growth response of each isolate as described previously [17,31]. The isolates were designated as resistant (>40% growth at 5 and 100 µg mL^−1^ metalaxyl), intermediate (>40% growth at 5 µg mL^−1^ metalaxyl) and sensitive (<40% growth at 5 and 100 µg mL^−1^ metalaxyl), respectively.

### 2.3. Aggressiveness Test on Commercial Potato Cultivars

Three *P. infestans* isolates EG 5 (reference 13_A2 strain) [13], EG 237 and EG 276 were selected to determine their virulence profile by inoculating 10 potato cultivars (Anabel, Cara, Diamont, Lady Balfour, Lady Rosetta, Mondial, Santana, Charlotte, Spunta and Valor). Potato seed tubers were grown in 25 cm plastic pots filled with a mixture of sterilized sand and peat moss (1:1, *v*/*v*) under greenhouse conditions. Temperature range was 15 to 25 °C with ~80% relative humidity. Leaflets were detached from 60-day-old potato plants. Leaflets and tuber slices were inoculated with 20 μL of sporangial suspension (approx. 150,000 sporangia mL^−1^) of each isolate and were maintained in ~80% relative humidity at 20 °C in transparent plastic boxes. Each treatment was replicated twice and the data were collected at 7 days post-inoculation. Leaflets were evaluated for the presence of sporulation and considered as virulent or avirulent [26] and the number of sporangia produced per leaflet/slice was counted using a haemocytometer [32].

### 2.4. DNA Extraction and Genotype Identification

The purified pure cultures of 152 Egyptian isolates and 15 reference isolates (previously reported by El-Ganainy et al. 2022 from Egypt) were grown on RAM. The mycelium was collected separately from each isolate for total genomic DNA extraction using DNeasy^®^ Plant Mini Kit (Qiagen, Hilden, Germany), according to the manufacturer’s protocol. Genotyping of the sampled and reference isolates was performed using a 12-plex PCR technique employing multiplexing of twelve SSR markers [10]. Markers used in this study were D13, G11, Pi4B, Pi04, Pi63, Pi70, PinfSSR2, PinfSSR3, PinfSSR4, PinfSSR6, PinfSSR8 and PinfSSR11 (Appendix A). One of the fluorescent dyes, FAM, VIC, NED or PET, was used to label one primer for each locus and the primers were prepared in dark cryotubes. All 24 primers were multiplexed in one PCR tube using the QIAGEN Multiplex PCR Kit (Qiagen, Germany) to prepare one PCR reaction for each DNA sample. PCR reactions were performed in a thermocycler (MWG-Biotech Ebersberg, Germany) [10]. The integrity and conformation of PCR amplicons were confirmed with agarose gel electrophoresis and an ABI3730 DNA analyzer (Applied Biosystems). The fragment size of SSR alleles was measured and scored using GeneMapper v3.7 software (Applied Biosystems). Detailed protocols are available at www.euroblight.net (accessed on 23 January 2023).

### 2.5. Screening AVR2 and AVR2-like Effector Genes

The presence and absence of the AVR2 and AVR2-like effector genes responsible for virulence/a virulence activity were confirmed in eleven selected isolates (Appendix A). DNA was extracted as above and PCR was performed in 25 µL reaction volume using primer pair PiAVR2_F2 (GACCAAACGGCGTACTTCAT)/PiAVR2_R2 (CGCCGAGCTCTTAACTCCT) for AVR2 and Piavr2_F7 (ACGCTTCTATCCGACAACGA)/Piavr2_R7 (ATTGGTGGTAATGCCTGCG) for AVR2-like effector genes [11]. The PCR master mix contained 1 µL of the DNA extract (~40 ng of total DNA), 2.5 µL AccuTaq LA 10X Buffer, 1.5 µL of 10 µM of each primer, 1 µL of DMSO, 1.25 µL dNTPs (10 mM each) and 0.25 µL AccuTaq™ LA DNA Polymerase (Sigma-aldrichD8045, Foster City, CA, USA) and nuclease-free water to make a final volume of 25 µL. PCR was conducted using the ESCO Swift Maxi Thermal Cycler and amplification conditions with an initial denaturation at 95 °C for 5 min, followed by 35 cycles of 95 °C for 1 min, 55 °C for 30 sec, 72 °C for 1 min and a final cycle of 72 °C for 10 min. The amplified PCR products were cleaned up and sequenced in both directions at Macrogen, Inc. (Seoul, South Korea). Obtained sequences were edited and aligned using MEGA11 [33], and single nucleotide polymorphisms (SNPs) were compared with the reference avirulent strain EG_40 (AVR2 accession No. MG976593 and AVR2-like accession No. MG976600).

### 2.6. Data Analysis

The number of sporangia per lesion were subjected to analysis of variance (ANOVA) and statistical analysis was performed separately for each isolate on tuber slices and leaflets of ten potato cultivars. Mean values of sporangia were compared using the Least Significant Difference (LSD) test at (*p* < 0.05) using SPSS software v. 8.0 [34]. The R packages POLYSAT and *poppr* v. 2.4.1 [35] were used for population genetic analysis. A principal component analysis (PCA) was calculated from Bruvo distances using POLYSAT and the distance matrix was exported from POLYSAT to PHYLIP to perform phylogenetic analysis. For *poppr* analysis, the basic parameters included the number of observed alleles (NA), number of multi-locus genotypes (MLGs), Stoddart and Taylor’s index of MLG diversity [36], Nei’s unbiased gene diversity, Hexp [37] and indices of evenness by locus [38]. Genotypic richness was calculated as R = (M − 1)/ (N − 1), where M is the number of different MLGs and N is the number of samples for each population [38]. Pairwise genetic distances were calculated using Nei’s formula [39] and a binary representation of the presence (1) or absence (0) of alleles. A minimum spanning network was generated based on the Bruvo genetic distances calculated using *poppr*. STRUCTURE v. 2.3.3 was used to infer a population structure based on the Bayesian clustering method described by Pritchard et al. [40]. The admixture model was applied with 20 runs for each K value from 2 to 10, each run with a burn-in period of 250,000 generations and 500,000 Markov chain Monte Carlo (MCMC) replications. The optimal K statistic was determined using STRUCTURE HARVESTER (https://taylor0.biology.ucla.edu/struct_harvest/ accessed on 21 December 2022), which is based on the method of Evanno et al. [41].

## 3. Results

### 3.1. Sample Collection and Isolation

A total of 152 isolates (82 from potato, 69 from tomato and one from eggplant) were collected during the winter season of 2013, 2014, 2016, and 2018 from nine Egyptian governorates (Figure 1A). The majority of the isolates (28.3%) were obtained from Menofia (43 of total 152 isolates) followed by 21.1% from Beheira (32 of total 152 isolates), 18.4% from Kafr El Sheikh (28 of 152 isolates), 8.6% from Giza (13 of 152 isolates) and the remaining 23.7% from Gharbia, Qalyubia, Fayoum, Dakahlia and Sharqia governorate (36 of 150 isolates), respectively (Figure 1B). Typical late blight symptoms were observed on tomato, potato and eggplant leaves (Figure 2A–C). Microscopic examination confirmed the structures as typical semi-papillate sporangia of *P. infestans* (Figure 2D,E). Surprisingly, typical late blight symptoms were also observed on eggplant seedlings (*Solanum melongena* L.) in a greenhouse in Beheira governorate in northern Egypt. Oospores were observed when the eggplant isolate of *P. infestans* interacted with A2 reference isolate on PAM (Figure 2F). Four isolates were obtained from potato cv. Bellini in seasons 2013, 2014 and 2018 and four isolates from potato cv. Cara. These cultivars were previously reported as resistant to clonal lineage 23_A1 in Egypt [42].

### 3.2. Mating Type and Metalaxyl Sensitivity Assessment

Mating type testing showed that all (152) collected *P. infestans* isolates were A1 mating type. Neither the mating type A2 or self-fertile (SF) isolates were recorded in the Egyptian population during the study seasons 2013, 2014, 2016 and 2018.

Metalaxyl sensitivity testing of the 152 isolates revealed that 53% of isolates (81 of 152) were sensitive, 32% of isolates (48 of 152 isolates) were intermediate, while only 15% of the isolates (23 of 152 isolates) were resistant. In the 2013 growing season, 81% of isolates were identified as metalaxyl sensitive. Nevertheless, the proportion of metalaxyl-sensitive isolates of *P. infestans* was dramatically reduced to 44%, 60% and 40% in the successive years of 2014, 2016 and 2018, respectively (Figure 3).

### 3.3. Aggressiveness Tests

Three isolates of *P. infestans* EG-005 (genotype 13_A2), EG-237 and EG-276 (genotype EU_23_A1) were selected to determine their sporulation capacity on tubers and detached leaflets of ten potato cultivars commercially grown for potato production in Egypt. The results showed that the isolates EG-276 (isolated from potato cv. Cara) and EG-237 (isolated from the eggplant) were the most aggressive isolates on potato tuber slices. Based on the tuber slice datasets, the most resistant potato cultivar was Cara, while Santana was the most susceptible cultivar. Surprisingly, low numbers of sporangia were observed on the tuber slices of Spunta, which is reported as the most susceptible commercial cultivar to *P. infestans* (Figure 4A).

It was noticed that the sporulation of EG-237 was less on leaflets of Cara and Valor (Figure 4B), as compared to tuber slices of both cultivars. Whereas, both EG-005 and EG-276 isolates showed similar sporulation capacity on tuber slices and leaflets of all tested cultivars. All three isolates produced 2 to 20 times more sporangia on leaflets compared to tuber slices but aerial mycelium was more abundant on tuber slices.

### 3.4. Genotyping of Egyptian Isolates

All purified 152 Egyptian and 15 reference isolates were subjected to genotyping using a 12-plex PCR method to determine the population structure of the Egyptian isolates (Appendix A). All twelve SSR markers, except Pi70, were polymorphic with allele diversity ranging from two (Pi04, Pi63, SSR2, SSR4, SSR6 and SSR8) to seven reported alleles (D13) (Table 1). Overall, 38 different alleles were identified with an average of 3.17 alleles per isolate across the 12 SSR loci (Table 1). The highest diversity was in locus D13 with seven alleles followed by PiG11 and PinfSSR4 loci with six and five alleles, respectively. The expected heterozygosity, Hexp [37] was highest for the locus PiG11 (0.74) and lowest for the locus SSR6 (0.006) with an average of 0.42 alleles per locus. An average index of evenness [38] of 0.799, with a maximum of one for SS8 and Pi63 and a minimum of 0.33 for SS6, indicates that alleles were evenly distributed (Table 1). The genotype accumulation curve showed a plateau from the 11th SSR marker, indicating that our results had significant power to describe genetic the variation in the Egyptian *P. infestans* population (Figure 5).

The SSR marker data showed that all 152 *P. infestans* isolates belonged to the lineage EU_23_A1 and 27 MLGs were discriminated within this lineage. The majority of isolates (68) were grouped into MLG_21 (named as sub-clonal lineage 23_A1_19 in Appendix A), followed by MLG_33 (29) and MLG_28 (14) as sub-clonal lineages 23_A1_12 and 23_A1_10, respectively. There were fewer than ten isolates in all other MLGs (Appendix A). The highest number of genotypes (24) was identified in 2014, followed by only 9, 2 and 3 genotypes in 2013, 2016 and 2018, respectively. Among the 27 MLGs, 15 were detected only once, of these 4 and 11 were identified in 2013 and 2014, respectively. Only the genotype MLG_21 was identified in all sampling years during this study. The genotypes MLG_28 and MLG_33 had been detected in Egypt from 2011. Five genotypes (MLG_14, MLG_15, MLG_21, MLG_28, and MLG_33) were identified in 2013 and 2014. The genotypes MLG_11 and MLG_17 were exclusively found in season 2014, while the genotype MLG_20 was exclusively found in season 2018 (Appendix A). No correlation was observed among the geographical location, cultivar specificity and metalaxyl sensitivity for the MLGs. The genetic relatedness between the MLGs was shown in the phylogenetic tree (Figure 6). Additionally, the sub-clones 23_A1_10 and 23_A1_12, were reported in this study, which have been detected since 2011. A total of 16 new sub-clones were detected in 2014 (107 of 152 isolates). Furthermore, many variants were detected in the two loci PiG11 and D13. The isolate EG-237 from the eggplant belonged to the sub-clone 23_A1_19, which was the most frequent sub-clone in 2013 (Figure 6A,B, Appendix A).

The previously reported 13_A2 clonal lineage could not be detected from any governorate during this time period. In 2013, four new sub-clonal lineages of 23_A1 were detected (23_A1_20, 23_A1_21 and 23_A1_22).

The SSR profiles of the 152 Egyptian isolates in this study were compared to the 15 reference isolates with a total of 167 isolates. In the phylogenetic tree (Figure 6), the PCA (Figure 7), the structure analysis (Figure 8) and the minimum spanning network (Figure 9) of the 23_A1 lineage were clearly distinct from 13_A2 samples that were not detected in this sample from Egypt.

In the phylogenetic tree based on the Bruvo distances, the relationships between the forty 23_A1 MLGs is visualized in relation to the reference samples of this lineage from previous studies. For example, MLG_28 (n = 14) was sampled in 2011 and again in 2013 as well as 2014 (Appendix A) clustered close to MLG_33 (n = 29), which was only a single stepwise mutation (G11 208bp to 2010bp) from it (Appendix A) and sampled in 2011, 2013 and 2016 (Figure 6A). The PCA analysis is an overview of the genotyped and previously reported Egyptian isolates (15 reference isolates). The PCA clearly showed that the only clonal lineage in the current study was 23_A1, which was clearly distinct from the previously found 13_A2 clonal lineage in Egypt (Figure 7).

To assess the population structure, the clustering of genotypes without prior information on the origin of samples was studied using STRUCTURE software. The structure analysis results clearly showed the distinction between 13_A2 and 23_A1 lineages (Figure 8). The 13_A2 isolates were found to be clonally pure as indicated by a single color (red). On the contrary, the 23_ A1 isolates were indicated by two different colors (green and blue), suggesting the multiple ancestries for isolates in the 23_A1 lineage.

The minimum spanning network (MSN) of all 167 Egyptian *Phytophthora* isolates shows the relatedness amongst the 164 isolates of 23_A1 (152 newly purified, 12 purified in the previous study) and how genetically distinct they are from the three 13_A2 isolates and one miscellaneous isolate (misc). Two dominant MLGs are apparent, from which a continual process of SSR mutation has created many closely related but novel variant types. The data of MSN support the phylogenetic and PCA analysis (Figure 9).

### 3.5. Analysis of the AV2 and AVR2-like Effector Genes

The eleven randomly selected isolates of *P. infestans* were tested using a PCR assay to detect the presence or absence of the AVR2 and AVR2-like RXLR effector genes. Both genes were successfully amplified from all eleven tested isolates (Table 2). The amplicons of both genes were sequenced to detect any single nucleotide polymorphisms (SNPs). The resultant sequences of both genes were subsequently deposited in NCBI GenBank and their accession numbers are reported in Table 2. The data showed that one heterozygous (non-synonymous) SNP was detected in the AVR2 gene of six isolates (EG-275, EG-276, EG-277, EG-278, EG-279 and EG-280) at the nucleotide position 93–31^aa^ in the N terminus (AAW-N/K) (Figure 10A). While, the remaining five isolates had a conserved sequences as previously reported in the reference isolate EG-40 (accession No. MG976593). Two forms of the AVR2-like effector gene were detected in the eleven *P. infestans* isolates. The first form contained two heterozygous (non-synonymous) SNPs: AYG/MT at nt position 29–10^aa^ in the N- terminus and RTC/IV at nt position 274–92^aa^ in the C- terminus. These two heterozygous SNPs in the AVR2-like were found in four isolates (EG-273, EG-277, EG-278 and EG-280) (Figure 10B).

## 4. Discussion

This study about the population structure of *P. infestans* in Egypt over four seasons has identified 23_A1 as the only clonal lineage in Egypt. The phenotypic data (mating type, metalaxyl response and aggressiveness) and genotypic data (SSR analysis, multi-locus genotyping and effector diversity) were analyzed to determine the overall population structure and to describe the emergence, dynamics and displacement of the main clonal lineages in Egypt.

All the collected isolates from tomato and potato crops in nine governorates belonged to the A1 mating type, showing a clear dominance in the population of *P. infestans* over the four sampling years. This finding corroborates the previous studies that the A1 mating type has been established dominantly in the Egyptian agro-ecological environment [13,24,29,43]. Surprisingly, no A2 or SF isolates could be sampled during this study. Phenylamide fungicides (metalaxyl, oxadixyl, benalaxyl) act through inhibition of ribosomal RNA synthesis in the pathogen cells [44]. Although the exact genetics of metalaxyl resistance is unknown, it is believed to be associated with multiple mutations in the *RPA190* gene [45] or the spontaneous deletion of chromosome regions containing the sensitive alleles of this gene [45,46]. However, studies on the inheritance in genetic crosses of *P. infestans* also suggest other genes are involved [46]. Unlike 13_A2, the 23_A1 genotype is considered as sensitive to intermediate in response to metalaxyl [47]. This was supported in our results, which showed the majority of 23_A1 isolates (53%) were sensitive. However, the application of metalaxyl places a selective pressure on *P. infestans* to survive and more resistant strains may emerge within a clonal lineage through a range of epigenetic processes such as gene deletion and silencing that do not involve specific mutations in the DNA sequence. Such resistance that is unrelated to a candidate resistance mutation in the *RPA190* gene has been reported in the 23_A1 lineage in the US population [45]. The virulence profile of three *P. infestans* isolates was tested. EG-005 isolated from potato and of the 13_A2 lineage was compared to two isolates of 23_A1; EG-237 was selected as the only isolate from Eggplant and EG-276 since it was isolated from potato cv. Cara has been reported as resistant to 23_ A1 lineage. The most aggressive isolate on average over all cultivars on tuber slices with high sporulation capacity was EG-237 followed by EG-276 and then the 13_A2 isolate EG-005. Although this is based on only one component of aggressiveness (sporulation), these results may explain why 23_A1 isolates spread and became prevalent while 13_A2 disappeared over this period in Egypt. Our data suggest that the previously reported virulence profile of 23_A1 has changed and highlights that cultivar resistance ratings may vary over time due to the dynamic and evolving population of *P. infestans* [19,26,48]. The isolates of lineage 13_A2 were highly aggressive and rapidly spreading as dominant genotypes in many European countries [18]. These isolates have been shown to break the existing resistance in some potato cultivars due to their aggressive phenotype; however, these have a short latent period [49]. Another possible reason behind the decline of the 13_A2 lineage in Egypt may be the replacement of phenylamide fungicide (metalaxyl) with other active ingredients that have reduced the selection pressure on the predominantly resistant 13_A2 lineage and thus supported the spread of 23_A1.

The genotyping of 152 isolates using a set of 12 highly informative SSR markers corroborated the mating type tests and revealed that all were of the 23_A1 lineage. Twenty-seven sub-clones were observed within 23_A1 lineage. Polymorphisms were observed in mainly 3 loci (G11, D13, Pi02) discriminating 27 sub-clonal lineages of 23_A1 in Egypt. The cluster analysis also grouped all the genotyped and reference isolates into two main clusters, i.e., 23_A1 and 13_A2. Moreover, although 23_A1 has already been reported during 2010–2012 [13], the presence and dominance of 23_A1 in Egypt throughout the sampling years indicated that there has been a shift in the Egyptian *P. infestans* population. Such a pathogen shift has already been reported in Italy [48]. The widespread distribution of the 23_A1 clonal lineage on both potato and tomato crops could be attributed to the adaption to Egyptian climatic conditions. It should be noted that a total of 15 isolates were obtained from infected potato and tomato crops in late April 2013, 2014 and 2018, while the temperature was more than 30 °C, which indicated that some isolates of 23_A1 lineage might have become tolerant to warmer Egyptian agro-ecological conditions [13]. In addition, this lineage has previously caused a late blight pandemic in potato and tomato cultivations in Tunisia and tolerance to high temperatures was reported [26,50].

Nine new sub-clones of 23_A1 lineage were detected in this study during the sampling year 2013. The newly detected sub-clonal lineages 23_A1_20 contained a tri-allelic combination at the D13 locus (136, 140, and 210 bp) along with an additional new allele combination at the PiG11 locus (146 bp). This is the first detection of a tri-allelic combination at the D13 locus within this lineage in the Egyptian population. The sub-clone 23_A1_21 also contained the same 146 bp allele at the PiG11 locus. The sub-clone 23_A1_22 contained three alleles at the PiG11 locus (146, 156, 210 bp) along with a 140 bp allele and a new allele at the D13 locus (212 bp). The sub-clone 23_A1_23 contained two alleles (268, 270 bp) at the SSR3 locus. Meanwhile, in the sampling year 2014, 16 new sub-clonal lineages of 23_A1 were detected. Moreover, the most frequent sub-clonal lineage 23_A1_19 in the year 2013 was detected from the eggplant sample in 2014. The emergence of sub-clonal lineages in the Egyptian 23_A1 lineage corroborates with the previous findings where sub-clonal diversity of 23_A1 have been reported from Algeria [19], southern Europe [48] and Tunisia [26].

The population structure of *P*. *infestans* is influenced by many factors such as migration [19], cultivars and late blight management systems [51], sexual recombination [52], competition and climate changes. The A1 mating type of *P. infestans* has been pre-dominating in Egypt at least since 1941 [23]. Until today, the way in which *P. infestans* has been introduced to Egypt was not known. Later on, the A2 mating type was migrated in association with potato tubers imported from Egypt to the UK [53]. Subsequently, its spread extended to Europe [54,55,56], and other countries including China and India [57,58], with increased the aggressiveness and virulence reported on commonly cultivated commercial cultivars [55,59]. Unexpectedly, the A2 mating type was not reported in this study among the sampled population over the monitored four years. However, this A2 mating type comprised 70% of the population during the 2009/2010, 2010/2011 and 2011/2012 seasons [13]. The previously dominant A2 mating type appears to have been displaced following the spread of an A1 mating type lineage over the monitored years.

More than 700 pathogenicity effector proteins have been found located in repeat-rich and dynamic regions of the genome of *P. infestans* [60]. These pathogenicity effectors encoded avirulence (AVR) genes, which interact with their corresponding R genes in the host plants [61]. Studying such molecular interactions are helpful to unravel the primary molecular mechanisms of resistance against late blight [62]. Similar to other oomycetes, the most studied effector AVR genes in *P. infestans* belong to the RxLR group due to the presence of a common RxLR domain [63]. Two commonly studied effector genes AVR2 and AVR2-like that related to resistance genes in the differential set were selected to study their presence or absence in eleven 23_A1 genotyped isolates from Egypt. Consistent with previous reports [11,13], both effector genes were successfully detected from all isolates. One non-synonymous SNP was identified in the AVR2 gene of six isolates in the N-terminus, where it modified the amino acid asparagine (N) to the amino acid lysine (K). As reported previously, the presence of this AVR gene results in an avirulent reaction on the R2 Blacks differential [26] but the K/N amino acid changes do not modify this avirulence. This is consistent with the avirulence of lineage EU_23_A1 against R2 [12]. In the AVR2-like gene, two SNPs were detected at the N- and C-terminus of the *P. infestans* isolates; however, the specific host R-gene(s) that this effector interacts with are not yet known. The DNA nucleotide SNPs and amino acid changes detected in the 23_A1 lineage in this study were identical to those detected in other European lineages of *P. infestans* examined previously [26], suggesting a degree of conservation in these effectors. It was unexpected that differences in the heterozygous SNPs were found within the clonal EU_23_A1 lineage and that there was no correlation between the isolates with SNPs in the AV2 and AVR2-like gene. Also worth noting is that these SNPs were detected only in the N-terminus of the effector genes and according to Gilroy et al. [11] the role of the N-terminus is not considered as critical in virulence or avirulence of *P. infestans* clonal lineages. Understanding such effector gene diversity and evolution is crucial in understanding the host-pathogen interactions to plan the future deployment of durable resistance genes as part of IPM strategies against late blight in potato and tomato [64].

## 5. Conclusions

The results of the current study provide important insights into the diversity of dynamic populations of *P. infestans* over four seasons. The sampled population comprised only isolates of the 23_A1 clonal lineage which documents a transition from a previously mixed population to one dominated by a single A1 lineage. A total of 27 sub-clones of 23_A1 were detected in this study. The abundance of the 23_A1 sub-clones suggests that this lineage has been present for several years in Egypt and has accumulated mutations. We reported the occurrence of 23_A1 lineage on eggplant but only at the early stage of crop development. Occurrence of late blight on such alternative hosts may lead to greater challenges to disease management through the evolution of aggressive strains with a short latent period and high sporulation capacity. The study highlights the potential for rapid change in the population structure and dynamics of *P. infestans* under the climate conditions of Egypt. Furthermore, 15% of *P. infestans* isolates were resistant to metalaxyl. This finding might be meaningful for growers and advisors who should be aware that fungicides containing metalaxyl may not be effective in managing field epidemics of potato late blight in Egypt and alternative fungicides must be considered. Furthermore, our survey indicated that the A2 mating type, which was dominant prior to 2012, has been largely replaced by the A1 mating type which limits the potential for sexual recombination in the population.

## Figures and Tables

**Figure 1 jof-09-00349-f001:**
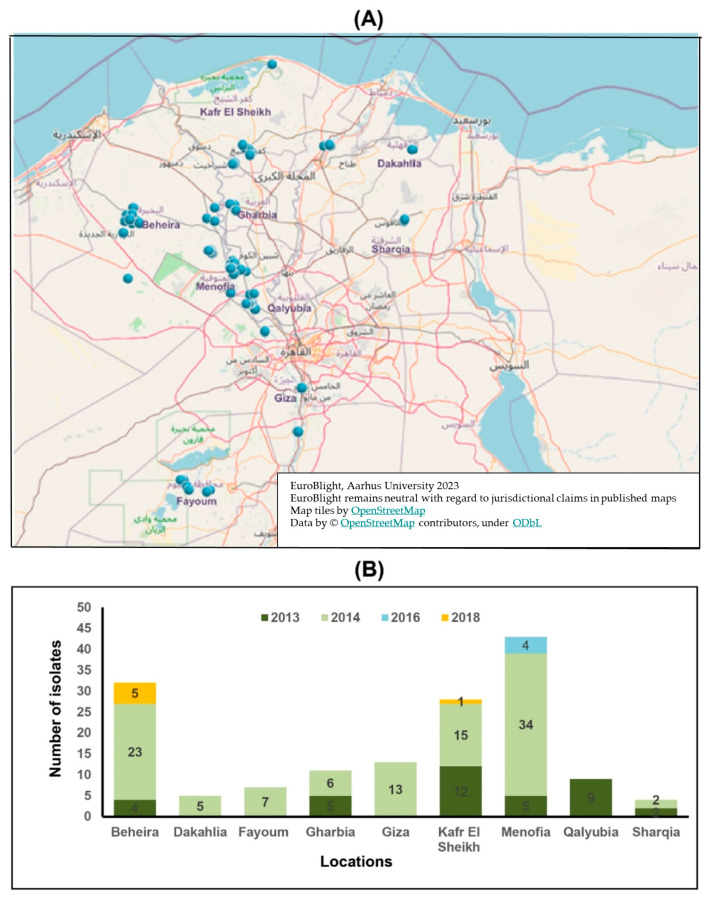
Late blight monitoring map showing locations of nine Egyptian governorates (**A**), Number of obtained isolates from nine different Egyptian governorates during four growing seasons (**B**). The map was drawn using online Euroblight database (https://agro.au.dk/forskning/internationale-platforme/euroblight/pathogen-monitoring/genotype-map, accessed on 21 January 2023).

**Figure 2 jof-09-00349-f002:**
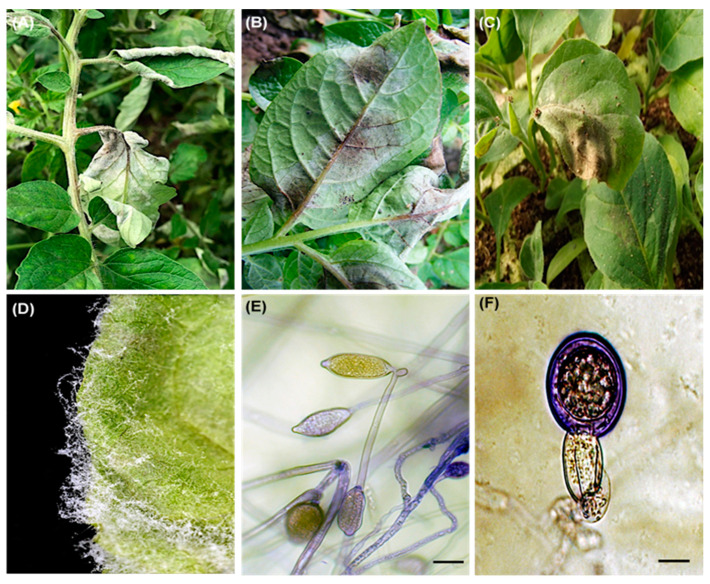
Late blight lesion on tomato plant showing white sporulation on the abaxial side of infected leaves (**A**), sporulated lesions on potato leaves (**B**), late blight symptoms on eggplant leaves (**C**), a close-up image of sporulating lesions and sporangia (**D**), microscopic view of semi-papillate sporangia (**E**), oospore inside an oogonium attached with an amphigynous antheridium (**F**). Scale bar 10 µm.

**Figure 3 jof-09-00349-f003:**
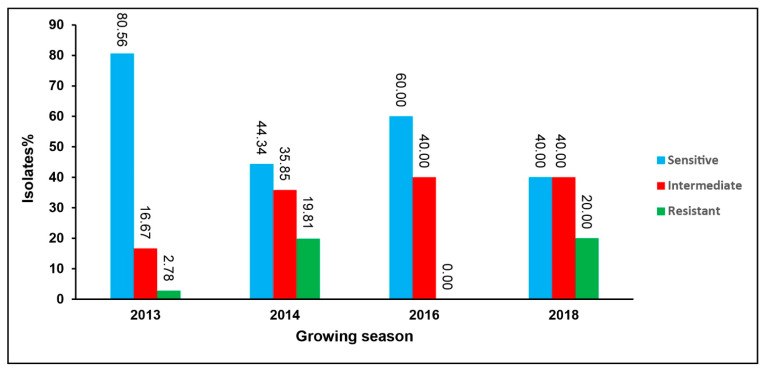
Metalaxyl sensitivity of isolates of *P. infestans* during 4 growing seasons (2013–2018) with 36, 106, 4 and 6 isolates tested, respectively.

**Figure 4 jof-09-00349-f004:**
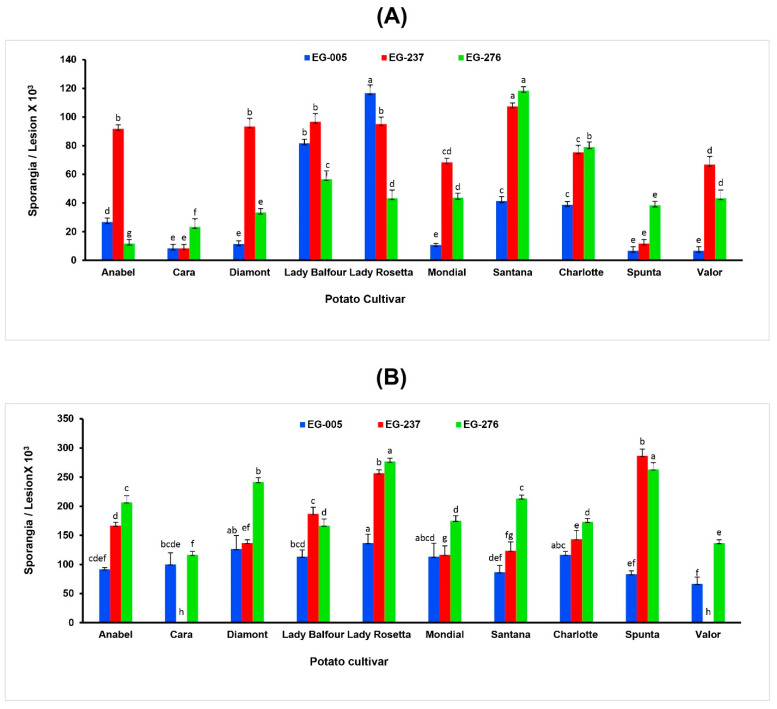
Number of sporangia (×10^3^) of *P. infestans* per lesion from (**A**) tuber slice and (**B**) detached leaflet infection assays on 10 potato cultivars. Columns bearing the same letters are not significantly different according to the Least Significant Difference (LSD) test (*p* < 0.05).

**Figure 5 jof-09-00349-f005:**
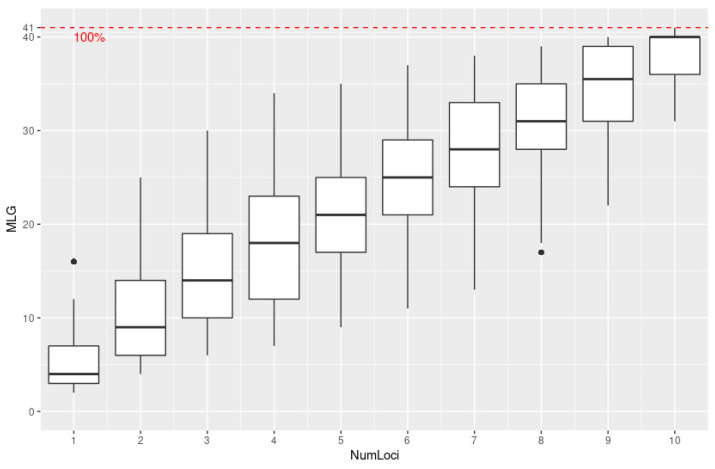
The genotype accumulation curve. Numloci is the number of SSR loci and MLG is the number of multi-locus genotypes. The resolution of MLG is 100% as shown in the dashed red line.

**Figure 6 jof-09-00349-f006:**
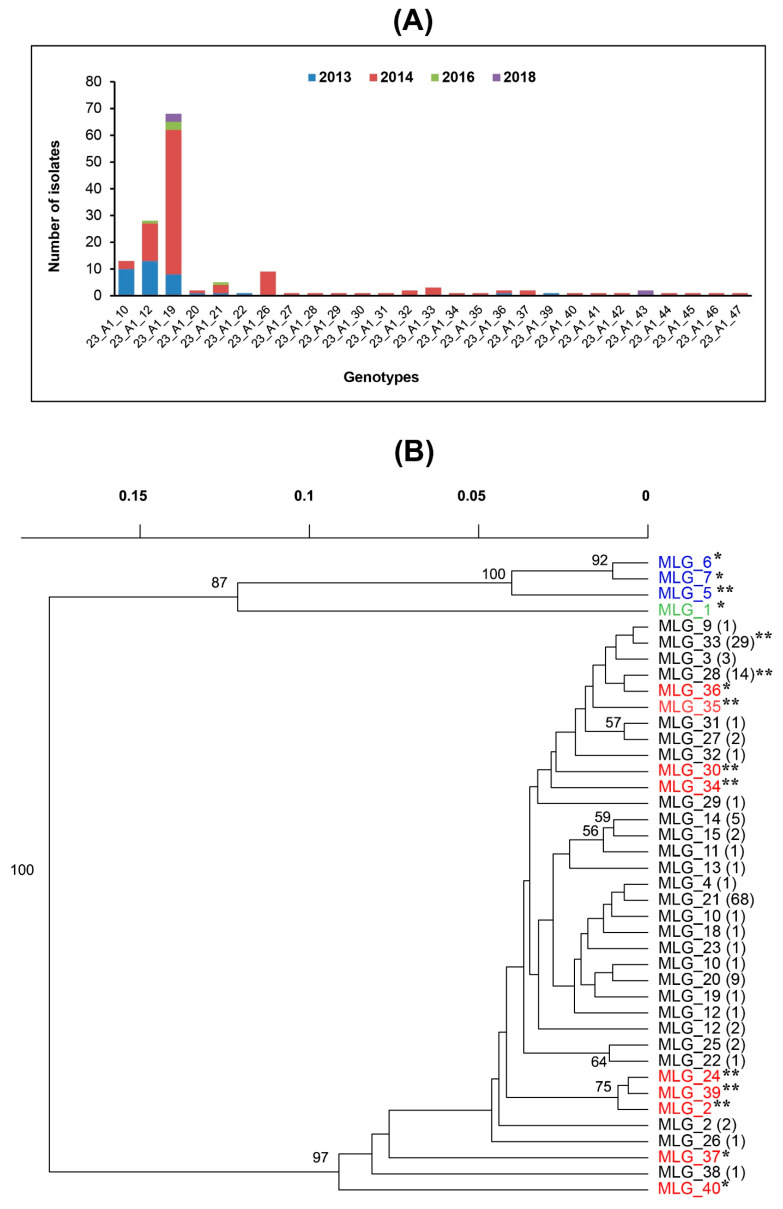
Number of isolates across Egyptian *P. infestans* populations according to MLGs, detected by 12 SSR markers. (**A**) The frequency of the 27 detected sub-clonal lineages over the 4 years. (**B**); Phylogenetic tree of the multi-locus genotypes (MLGs) of 23_A1 sampled in this study in comparison to reference isolates of 23_A1 (in red), the clonal lineage 13_A2 (blue) and a single isolate of another MLG of no known clonal lineage (green). The 23_A1 reference samples from previous work in Egypt are in red with a single asterisk representing isolates from 2010 and a double asterisk for subisolates from 2011. The numbers in brackets refer to the number of samples of each MLG.

**Figure 7 jof-09-00349-f007:**
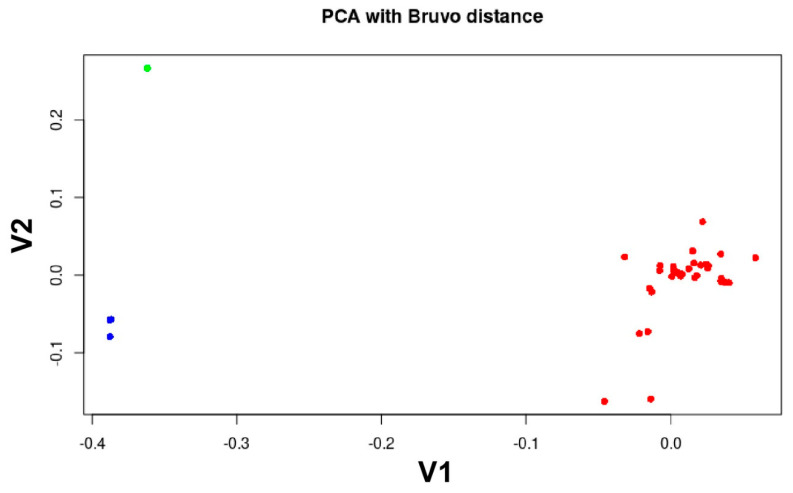
The principal components analysis (PCA) represents 167 selected isolates that were characterized using 12 SSR markers and examined in POLYSAT. PCA analysis revealed three major clusters: 13_A2 (represented in blue), 23_A1 (red) and unique lineage (green).

**Figure 8 jof-09-00349-f008:**
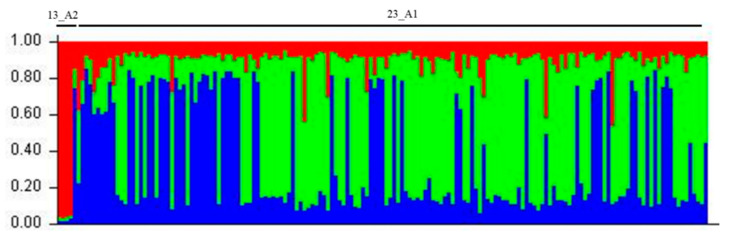
Genetic population structure of 167 Egyptian isolates of *P. infestans* with 12 microsatellite (SSR) markers in 2009–2016. Each color represents one cluster defined by STRUCTURE (Pritchard et al., 2000 [40]). The height of each color represents the probability of membership to each cluster estimated by STRUCTURE. Red color represents 13_A2 lineage. Blue and green color represent two possible clusters within the clonal lineage 23_A1, which does not correlate with geographical locations. The abbreviated genotype is given at the top.

**Figure 9 jof-09-00349-f009:**
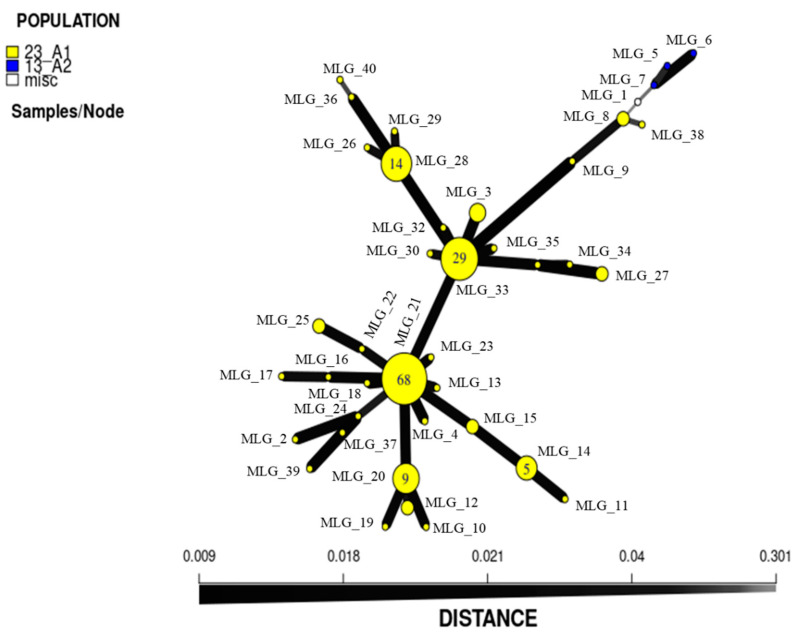
Minimum spanning network based on the Bruvo’s distance of the 167 *P. infestans* isolates using microsatellite (SSR) genotyping data. Nodes represent each multi-locus genotype, with the number of isolates indicated in the center of each circle. Lines connecting nodes with the lowest genetic distance are shown with thicker lines and the highest genetic distance is shown with thinner lines.

**Figure 10 jof-09-00349-f010:**
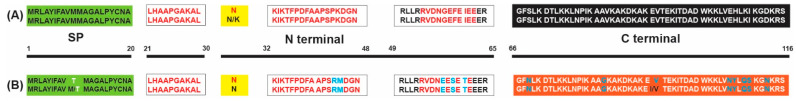
Amino acids alignment of the two effector genes AVR2 (**A**) and AVR2-like (**B**). The numbers above the black line show the start and end positions for each domain.

**Table 1 jof-09-00349-t001:** Summary of microsatellite (SSR) marker information and locus statistics of 152 *Phytophthora infestans* isolates in Egypt in 2013–2018 and 15 reference isolates.

SSR Locus	Allele Fragments	1-D ^b^	Hexp ^c^	Evenness ^d^
PiG11	6	0.7403	0.7419	0.8803
Pi02	4	0.6681	0.6696	0.9868
PinfSSR11	2	0.4998	0.5015	0.9996
D13	7	0.5467	0.5485	0.8023
PinfSSR8	2	0.5	0.5016	1
PinfSSR4	5	0.5501	0.5518	0.8347
Pi04	2	0.0131	0.0131	0.3278
Pi70	1	.	.	
PinfSSR6	2	0.0066	0.0066	0.2955
Pi63	2	0.5	0.5016	1
PinfSSR2	2	0.4998	0.5015	0.9996
Pi4B	3	0.5096	0.5113	0.9459

^b^ 1−D, Simpson’s diversity index [38]. ^c^
*H*exp, Nei’s gene diversity [37]. ^d^ Evenness, indices of evenness ([38]).

**Table 2 jof-09-00349-t002:** Accession numbers and details of 11 isolates of *P. infestans* and their host.

Isolate Name	Host	Location	MLG	Genotype	GenBank Accession Numbers	
					AVR2	AVR2-like
EG_ 237	*Solanum* *melongena*	Beheira	MLG_21	23_A1_19	OQ107577	OQ107598
EG_ 271	*S. tuberosum*	Menofia	MLG_33	23_A1_12	OQ107578	OQ107592
EG_ 272	*S. tuberosum*	Menofia	MLG_21	23_A1_19	OQ107579	OQ107594
EG_ 273	*S. tuberosum*	Menofia	MLG_21	23_A1_19	OQ107580	OQ107591
EG_ 274	*S. tuberosum*	Menofia	MLG_21	23_A1_19	OQ107581	OQ107593
EG_ 275	*S. tuberosum*	Beheira	MLG_14	23_A1_21	OQ107583	OQ107595
EG_ 276	*S. tuberosum*	Beheira	MLG_25	23_A1_43	OQ107584	OQ107596
EG_ 277	*S. tuberosum*	Beheira	MLG_21	23_A1_19	OQ107585	OQ107588
EG_ 278	*S. tuberosum*	Beheira	MLG_21	23_A1_19	OQ107586	OQ107589
EG_ 279	*S. lycopersicum*	Kafr El Sheikh	MLG_21	23_A1_19	OQ107582	OQ107597
EG_ 280	*S. tuberosum*	Beheira	MLG_25	23_A1_43	OQ107587	OQ107590

## Data Availability

All the data related to this study are mentioned in the manuscript and Appendix A.

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
