# Peer review of "Population Dynamics of Phytophthora infestans in Egypt Reveals Clonal Dominance of 23_A1 and Displacement of 13_A2 Clonal Lineage"

_jof, 2023, doi:10.3390/jof9030349_

Round 1
Reviewer 1 Report
The manuscript is devoted to the current topic. It includes studies of various isolates of the phytopathogenic fungus P. infestans isolated from Solanaceae growing in different regions of Egypt. The work uses modern methods. Well done statistical and bioinformatic analysis.
There are a number of remarks:
1. Too much abstract. It needs to be reduced.
2. How can you explain the resistance of some of the isolates you isolated to the fungicide metalaxyl? What is the mechanism of this stability? Why is the number of resistant isolates growing in dynamics, what is the reason for this? You write that this is a monogenic trait (lines 470-471), so explain what it is connected with.
3. Explain why the three isolates of P. infestans EG-005, EG-237 and EG-276 were chosen for the aggressiveness test. On what basis were the isolates selected?
4. What explains the highest number of isolates in 2014 (Fig. 6)?
5. In the discussion (lines 458-460) you write about the adaptation of fungal lines to climatic conditions. Please write in more detail how the climate in Egypt changed during the period under study and how P. infestans adapted to it. This is important for the reader to understand the meaning of your work.
6. In the Conclusion, you need to add more conclusions from your work, remove common phrases.
7. Very outdated References. Less than 10% of literary sources in it for the last 3 years. The bibliography needs to be updated.
Respectfully Yours, reviewer.
February 26, 2023
Reviewer 2 Report
The article in general is well written and the information presented in a logical manner but some information in sections Materials and Methods and Results must be completed.
I have made several queries throughout the manuscript, most of them minor.
